# Sub-50 nm perovskite-type tantalum-based oxynitride single crystals with enhanced photoactivity for water splitting

Jiadong Xiao [1], Mamiko Nakabayashi [2], Takashi Hisatomi [1], Junie Jhon M. Vequizo [1], Wenpeng Li[1], Kaihong Chen [1], Xiaoping Tao[1], Akira Yamakata[3], Naoya Shibata [2], Tsuyoshi Takata[1], Yasunobu Inoue[4] & Kazunari Domen [1,5] ✉

A long-standing trade-off exists between improving crystallinity and minimizing particle size in the synthesis of perovskite-type transition-metal oxynitride photocatalysts via the thermal nitridation of commonly used metal oxide and carbonate precursors. Here, we overcome this limitation to fabricate $A$TaO$_2$N ($A$ = Sr, Ca, Ba) single nanocrystals with particle sizes of several tens of nanometers, excellent crystallinity and tunable long-wavelength response via thermal nitridation of mixtures of tantalum disulfide, metal hydroxides ($A$(OH)$_2$), and molten-salt fluxes (e.g., SrCl$_2$) as precursors. The SrTaO$_2$N nanocrystals modified with a tailored Ir–Pt alloy@Cr$_2$O$_3$ cocatalyst evolved H$_2$ around two orders of magnitude more efficiently than the previously reported SrTaO$_2$N photocatalysts, with a record solar-to-hydrogen energy conversion efficiency of 0.15% for SrTaO$_2$N in Z-scheme water splitting. Our findings enable the synthesis of perovskite-type transition-metal oxynitride nanocrystals by thermal nitridation and pave the way for manufacturing advanced long-wavelength-responsive particulate photocatalysts for efficient solar energy conversion.

Perovskite-type oxynitrides of early transition metals and alkaline-earth or rare-earth elements, in optimal cases, can combine the advantages of oxides and nitrides, exhibiting greater stability in air and moisture than pure nitrides and narrower bandgaps than comparable oxides[1,2]. In recent decades, this superiority has made these perovskite-type compounds an important class of functional materials used in nontoxic pigments[3], colossal magnetoresistors[4], high-permittivity dielectrics[5], and long-wavelength-responsive photocatalysts[6]. In response to an urgent worldwide need to reduce carbon emissions to mitigate global warming, perovskite oxynitride semiconductor-based particulate photocatalysts have attracted particular attention because they potentially

enable the direct synthesis of sustainable fuels and chemical products using sunlight as the sole energy source[6–9]. The tantalum oxynitride perovskites, $A$TaO$_2$N ($A$ = Ca, Sr, Ba), are rare semiconducting materials with excellent stability in aqueous solutions, narrow bandgaps (1.9–2.4 eV), and conduction and valence bands appropriately positioned to straddle the water redox potential[6,10]. As such, they are regarded as among the most promising photocatalyst materials for overall water splitting (OWS), which can be used in particulate-photocatalyst-sheet-based panel systems for large-scale solar H$_2$ production[6,11].

Nitridation of oxide (with carbonate) precursors in flowing ammonia (NH$_3$) at high temperatures is the most widely applied

[1]Research Initiative for Supra-Materials, Interdisciplinary Cluster for Cutting Edge Research, Shinshu University, Nagano-shi, Nagano 380-8553, Japan. [2]Institute of Engineering Innovation, School of Engineering, The University of Tokyo, 2-11-16, Yayoi, Bunkyo-ku, Tokyo 113-8656, Japan. [3]Graduate School of Natural Science and Technology, Okayama University, 3-1-1 Tsushimanaka, Kita-ku, Okayama 700-8530, Japan. [4]Japan Technological Research Association of Artificial Photosynthetic Chemical Process (ARPChem), 2-11-16 Yayoi, Bunkyo-ku, Tokyo 113-8656, Japan. [5]Office of University Professors, The University of Tokyo, 2-11-16 Yayoi, Bunkyo-ku, Tokyo 113-8656, Japan. ✉e-mail: domen@shinshu-u.ac.jp

method for the synthesis of perovskite oxynitrides[2]. Such oxynitrides, including $A\text{TaO}_2\text{N}$ ($A$ = Sr, Ca, Ba) produced by thermal nitridation of polymetallic oxides, are generally polycrystalline and incorporate structural defects that act as recombination and trapping centers for photogenerated charge carriers[12,13]. This disadvantage can be overcome by using a mixture of polymetallic oxides (or metal oxides and carbonates) together with a molten-salt flux (e.g., NaCl, KCl, or RbCl) as the nitridation precursor[14–17]. The best-in-class $A\text{TaO}_2\text{N}$ semiconductors are prepared by this approach and are characterized by well-crystallized single crystals and improved photocatalytic activity[15–17]. However, the sizes of the resultant particles are uncontrollable and inevitably large (several hundred nanometers at minimum)[15–17], leading to long distances for the photoexcited charge carriers to migrate to reach active sites on the surface. This problem points to a long-standing trade-off between improving crystallinity and minimizing particle size in the synthesis of metal oxynitride perovskites via the flux-assisted thermal nitridation process. Overcoming this limitation would represent a critical advancement in long-wavelength-responsive perovskite oxynitride-based photocatalyst manufacturing yet remains a grand challenge.

Here, we show that using a mixture of tantalum(IV) sulfide ($\text{TaS}_2$), metal hydroxide ($A(\text{OH})_2$, A = Ca, Sr, Ba), and a molten salt (e.g., $\text{SrCl}_2$) as the nitridation precursor enables the fabrication of highly crystalline $A\text{TaO}_2\text{N}$ ($A$ = Ca, Sr, Ba) single nanocrystals with sub-50 nm particle sizes and a tunable long-wavelength response. Each precursor material was found to be critical to the formation of single nanocrystals that simultaneously exhibit a high degree of crystallinity and a small

particle size of a few tens of nanometers. This approach leads to high photocatalytic efficiency of the resultant $\text{SrTaO}_2\text{N}$ nanocrystals toward sacrificial $\text{H}_2$ and $\text{O}_2$ evolution and Z-scheme water splitting. In particular, Ir–Pt alloy@$\text{Cr}_2\text{O}_3$-modified $\text{SrTaO}_2\text{N}$ nanocrystals evolved $\text{H}_2$ approximately two orders of magnitude more efficiently than the previously reported $\text{SrTaO}_2\text{N}$ photocatalysts and were used, for the first time, as a $\text{H}_2$-evolution photocatalyst (HEP) for Z-scheme water splitting, providing a solar-to-hydrogen (STH) energy conversion efficiency of 0.15%.

## Results and discussion
### Synthesis and characterization of $A\text{TaO}_2\text{N}$ ($A$ = Sr, Ca, Ba) nanocrystals

Nitridation of a powder containing $\text{TaS}_2$ (Supplementary Fig. 1), $\text{Sr(OH)}_2$, and $\text{SrCl}_2$ in a molar ratio of 1:2.5:1 under a flow of gaseous $\text{NH}_3$ at 1223 K for 3 h yielded an orange powder (see details in Methods). The X-ray diffraction (XRD) pattern for the obtained powder indicated a single phase associated with perovskite-type $\text{SrTaO}_2\text{N}$ (Fig. 1a). The product also exhibited a light-absorption edge at approximately 600 nm, characteristic of $\text{SrTaO}_2\text{N}$ (Fig. 1b), and an average particle size of approximately 50 nm (Fig. 1c, d). Lattices fringes with the same orientation (Supplementary Fig. 2a) and a well-defined spot-like selected-area electron diffraction (SAED) pattern (Supplementary Fig. 2b) for a single particle, together with highly ordered lattice fringes at the outermost surface (Supplementary Fig. 3), were observed when a cross-sectional specimen of this material was examined by high-resolution transmission electron microscopy

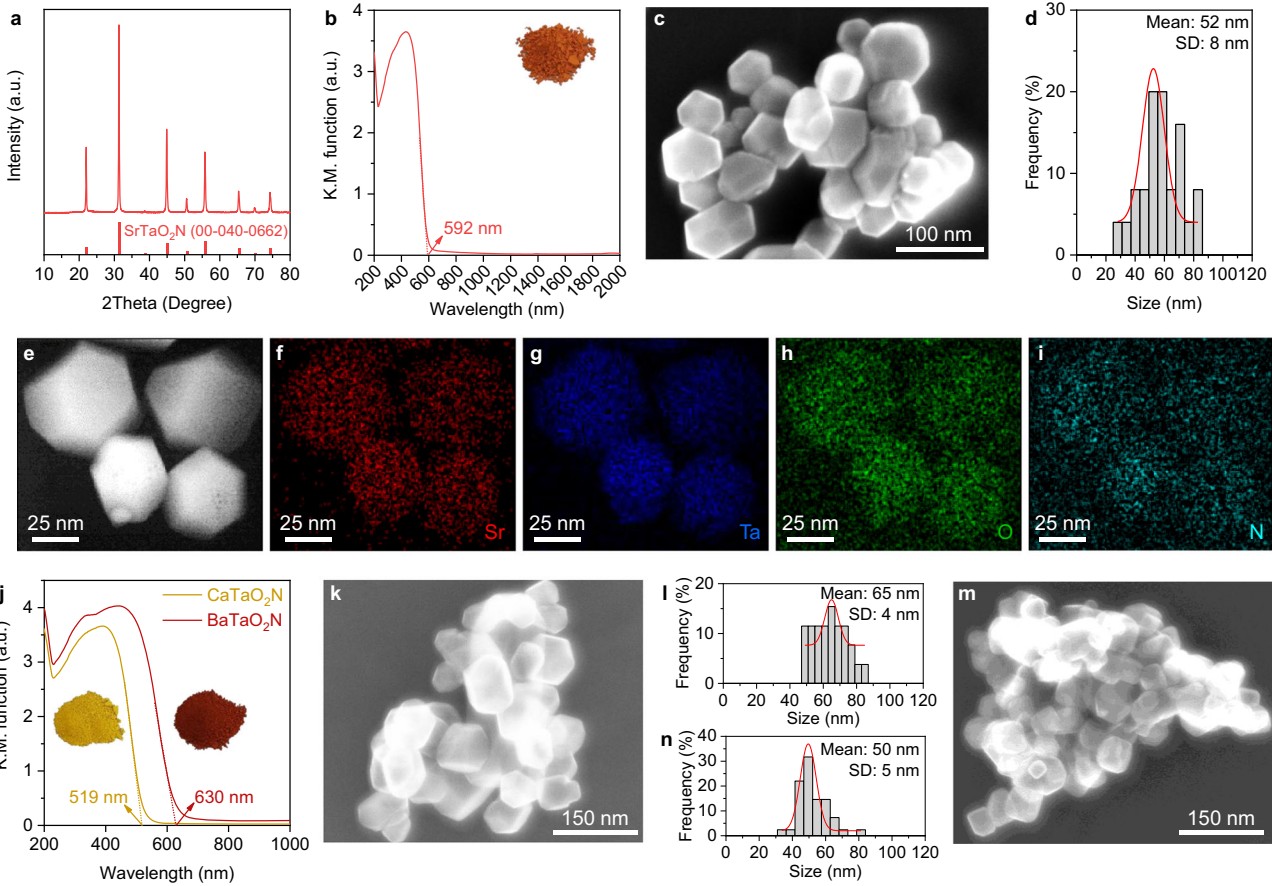

**Fig. 1 | Synthesis and characterization of nanocrystalline $A\text{TaO}_2\text{N}$ ($A$ = Sr, Ca, Ba).** XRD pattern (**a**), UV–vis diffuse-reflectance spectrum (**b**), Scanning electron microscopy image (**c**), and particle size distribution (**d**) of the synthesized $\text{SrTaO}_2\text{N}$. ADF–STEM image (**e**) and STEM–EDS elemental mapping images of Sr (**f**), Ta (**g**), O (**h**), and N (**i**) of a cross-sectional $\text{SrTaO}_2\text{N}$ sample. **j** UV–vis diffuse-reflectance spectra of the synthesized $\text{CaTaO}_2\text{N}$ and $\text{BaTaO}_2\text{N}$. SEM image (**k**) and particle size distribution (**l**) of $\text{CaTaO}_2\text{N}$. SEM image (**m**) and particle size distribution (**n**) of $\text{BaTaO}_2\text{N}$. The mean value and standard deviation (SD) of the particle sizes in (**d**, **l**, **n**) were determined by Gaussian fitting (red lines). The insets in (**b**, **j**) show photographs of the respective materials.

(HRTEM). On the basis of an elemental analysis, the chemical formula for the material was estimated to be $Sr_{1.00}Ta_{1.01}O_{2.07}N_{1.03}$ (Supplementary Table 1); the sulfur content (0.004 wt%) was below the detection limit (0.01 wt%). In keeping with this result, elemental mapping of a cross-sectional $SrTaO_2N$ sample using annular dark-field scanning transmission electron microscopy coupled with energy dispersive X-ray spectroscopy (ADF STEM–EDS) (Fig. 1e–i) indicated that Sr, Ta, O, and N were evenly distributed within single particles, whereas negligible sulfur was detected (Supplementary Fig. 4). Notably, the $SrTaO_2N$ nanocrystals exhibited extremely weak background absorption at longer wavelengths (>600 nm) (Fig. 1b), suggesting a low defect density[18,19]. These results indicate the formation of highly crystalline $SrTaO_2N$ single nanocrystals.

We found that each precursor component was critical to the formation of single nanocrystals of $SrTaO_2N$ with high crystallinity, and that the optimized $TaS_2/Sr(OH)_2/SrCl_2$ molar ratio in the precursor was 1/2.5/1 (see details in Supplementary Figs. 5–7). A several-micron-sized 2H-$TaS_2$ phase (Supplementary Fig. 1) has not been previously used for synthesizing perovskite-type oxynitrides; it was found to play an irreplaceable role in the developed approach. Nitridation of the $Ta_2O_5$/$Sr(OH)_2/SrCl_2$ (molar ratio: 0.5/2.5/1) mixture, where $TaS_2$ was replaced with $Ta_2O_5$ (the most widely used Ta source for synthesizing oxynitrides[2,14,15]), did not produce $SrTaO_2N$ when reacted under the same conditions but generated predominantly $Sr_6Ta_2O_{10.188}$ (Supplementary Fig. 5). Replacement of $Sr(OH)_2$ with $SrCO_3$ resulted in $SrTaO_2N$ nanoparticles but with relatively larger sizes (Supplementary Fig. 8). The molten-salt-assisted fragmentation of $TaS_2$ under the nitridation conditions is presumably a key reason for the formation of $SrTaO_2N$ nanocrystals (see detailed discussion below Supplementary Fig. 9). The $SrCl_2$ used as a molten-salt flux could be replaced with another congeneric flux. For instance, the use of NaCl, KCl, or RbCl instead of $SrCl_2$ led to $SrTaO_2N$ nanocrystals with an average particle size of approximately 20 nm (Supplementary Figs. 10 and 11). However, the crystallinity of the resultant $SrTaO_2N$ samples was lower, as indicated by the large increase in the full-width at half-maximum (FWHM) of their characteristic XRD peaks, associated with a decrease in the melting point of the molten-salt flux (see explanation below Supplementary Fig. 10).

We readily extended the proposed approach to the synthesis of $CaTaO_2N$ and $BaTaO_2N$ nanocrystals by simply replacing $Sr(OH)_2$ in the precursor with $Ca(OH)_2$ and $Ba(OH)_2$, respectively (Supplementary Fig. 12). The obtained yellow-colored $CaTaO_2N$ and crimson-colored $BaTaO_2N$ (Fig. 1j) exhibited average particle sizes of ~65 (Fig. 1k, l) and 50 nm (Fig. 1m, n), respectively. The light-absorption edges for the prepared $CaTaO_2N$ and $BaTaO_2N$ were located at 519 and 630 nm (Fig. 1j), respectively, whereas the typical values are 500 nm for $CaTaO_2N$ and 650 nm for $BaTaO_2N$[6]. This difference is attributable to the slight substitution of $Ca^{2+}$ and $Ba^{2+}$ in the oxynitrides by $Sr^{2+}$ as a result of the use of the molten-$SrCl_2$ flux. Accordingly, the XRD peak positions for the $CaTaO_2N$ and $BaTaO_2N$ samples were shifted to lower and higher angles, respectively (Supplementary Fig. 12). These results demonstrate that nitridation of mixtures of $TaS_2$, metal hydroxides, and molten-salt fluxes can be a universal and flexible approach for fabricating perovskite-type tantalum oxynitride nanocrystals with tunable light-absorption edge wavelengths.

## $H_2$-evolution and Z-scheme water-splitting performance of the $SrTaO_2N$-nanocrystal photocatalyst

As a demonstration, the $H_2$-evolution activity of $SrTaO_2N$ nanocrystals was evaluated after the nanocrystals were sequentially modified with Ir and Pt by microwave heating in water and ethylene glycol (EG), respectively, and finally with $Cr_2O_3$ by photodeposition (see details in Methods). The obtained photocatalyst (denoted as $Cr_2O_3$/Pt ($MW_{EG}$)/Ir ($MW_{H2O}$)/$SrTaO_2N$ nanocrystals) was found to evolve $H_2$ efficiently from an aqueous methanol solution (solid circles in Fig. 2a) in which

the feeding concentrations of Ir, Pt, and Cr (relative to the mass of $SrTaO_2N$) were optimized to be 0.5, 1.0, and 0.5 wt%, respectively (Supplementary Fig. 13). The $H_2$-evolution rate for the $Cr_2O_3$/Pt ($MW_{EG}$)/Ir ($MW_{H2O}$)/$SrTaO_2N$ nanocrystals was approximately five times higher than that for the $SrTaO_2N$ modified with the same cocatalysts but synthesized by nitridation of a typical $Ta_2O_5$/$SrCO_3$/$SrCl_2$ precursor and exhibiting an average particle size of 200 nm (open squares in Fig. 2a; see characterization results in Supplementary Fig. 14). Moreover, the single-nanocrystal $SrTaO_2N$ evolved $H_2$ four times higher than the polycrystalline $SrTaO_2N$ exhibiting aggregates composed of several polycrystalline nanoparticles with an average size of 63 nm (see detailed discussion in Supplementary Fig. 15) and 2.4 times higher than the $SrTaO_2N$ previously developed from a $Ta_2O_5$/$NaOH$/$SrCl_2$ precursor[17] (Supplementary Fig. 16). These results demonstrate the importance of both a small particle size and a high degree of crystallinity to the photocatalytic performance and the superiority of the developed approach in producing highly crystalline single nanocrystals of $ATaO_2N$ (A = Sr, Ca, Ba). The onset irradiation wavelength for $H_2$ generation agreed with the absorption edge for this $SrTaO_2N$ photocatalyst (Fig. 2b), indicating that the photoreaction proceeded via bandgap transitions. The associated apparent quantum yield (AQY) for $H_2$ evolution was calculated to be 3.0% at 422 nm, 2.6% at 479 nm, and 0.5% at 580 nm (Fig. 2b). Notably, the previously reported $SrTaO_2N$ photocatalysts evolve $H_2$ inefficiently with a rate well below 5 μmol h$^{-1}$, and the associated AQY is too low to be detected (Supplementary Table 2). An estimation based on the evolution rates for $H_2$ indicates that the developed $Cr_2O_3$/Pt ($MW_{EG}$)/Ir ($MW_{H2O}$)/$SrTaO_2N$ nanocrystal photocatalyst evolves $H_2$ approximately two orders of magnitude more efficiently than the previously reported $SrTaO_2N$ photocatalysts (Supplementary Table 2).

When the $H_2$-evolving $Cr_2O_3$/Pt ($MW_{EG}$)/Ir ($MW_{H2O}$)/$SrTaO_2N$ nanocrystals were combined with the reported Ir and $FeCoO_x$ nanocomposite co-modified $BiVO_4$ (Ir-$FeCoO_x$/$BiVO_4$)[20] (see characterization results in Supplementary Fig. 17) as the $O_2$-evolution photocatalyst (OEP) and $[Fe(CN)_6]^{3-}$/$[Fe(CN)_6]^{4-}$ as a redox mediator, both $H_2$ and $O_2$ were stably evolved under simulated sunlight at a near-stoichiometric molar ratio of 2:1 for 33 h with negligible deactivation (Fig. 2c). Upon modification with the same $Cr_2O_3$/Pt ($MW_{EG}$)/Ir ($MW_{H2O}$) cocatalyst, the $SrTaO_2N$ nanocrystals (solid circles in Fig. 2c) exhibited notably higher photocatalytic activity in Z-scheme water splitting than the large-sized $SrTaO_2N$ prepared from the typical $Ta_2O_5$/$SrCO_3$/$SrCl_2$ precursor (open squares in Fig. 2c). The STH energy conversion efficiency for this redox-mediated Z-scheme system was 0.15%, and the AQY was measured to be 4.0% at approximately 420 nm (Supplementary Fig. 18). These STH and AQY values are still lower than those for Z-scheme systems constructed with $Rh_yCr_{2-y}O_3$-loaded $ZrO_2$-modified TaON (AQY of 12.3% at 420 nm and STH efficiency of 0.6%)[20] or Ru-modified $SrTiO_3$:La,Rh (AQY of 33% at 419 nm and STH of 1.1%)[21] as HEPs. Nevertheless, this AQY is comparable to that for the Z-scheme system constructed with Pt/$BaTaO_2N$ as the HEP[15], both representing the most efficient bias-free Z-scheme water-splitting systems involving 600-nm-class photocatalysts. More importantly, our proposed Z-scheme system is, to the best of our knowledge, the first involving $SrTaO_2N$ as the HEP because previous $SrTaO_2N$ photocatalysts evolved $H_2$ inefficiently (Supplementary Table 2). Further improvements are expected by refining the operating parameters and exploring long-wavelength-responsive OEPs and more effective redox mediators or solid conductive mediators.

## Nanostructure and promotion effects of $Cr_2O_3$/Pt ($MW_{EG}$)/Ir ($MW_{H2O}$) cocatalyst

The substantial improvement in the $H_2$-evolution and Z-scheme water-splitting activity of the $SrTaO_2N$ nanocrystals also relied on the multicomponent $Cr_2O_3$/Pt ($MW_{EG}$)/Ir ($MW_{H2O}$) cocatalyst that can accelerate the surface $H_2$-evolution reactions much more efficiently than Pt,

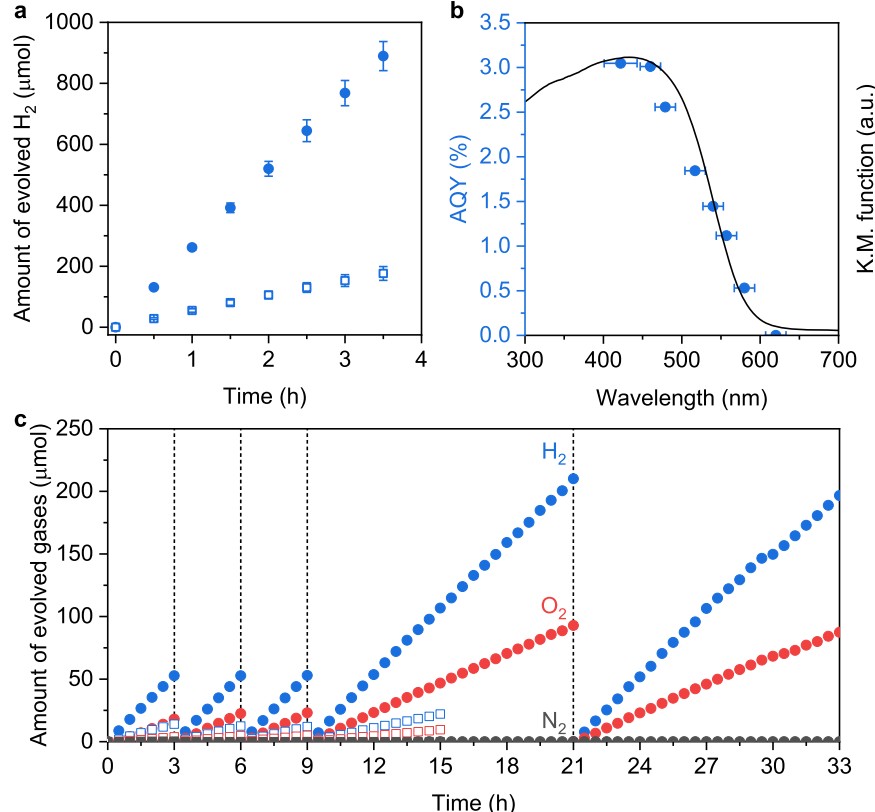

**Fig. 2 | H₂-evolution and Z-scheme OWS performance of SrTaO₂N nanocrystal photocatalyst. a** Time courses of the photocatalytic H₂-evolution reaction in aqueous methanol (13 vol%) solution over the Cr₂O₃/Pt (MW_EG)/Ir (MW_H2O)-modified SrTaO₂N nanocrystals (solid circles) and over the Cr₂O₃/Pt (MW_EG)/Ir (MW_H2O)-modified large-sized SrTaO₂N crystals (Supplementary Fig. 13) prepared from the Ta₂O₅/SrCO₃/SrCl₂ precursor (open squares). Error bars indicate the standard deviation of three measurements. **b** AQY as a function of the incident-light wavelength during visible-light-driven H₂ production over Cr₂O₃/Pt (MW_EG)/Ir (MW_H2O)-modified SrTaO₂N nanocrystals. The solid line indicates the UV–vis diffuse-reflectance spectrum of SrTaO₂N. The above reactions were carried out under illumination from a Xe lamp (300 W, $\lambda \geq 420$ nm) with or without various bandpass filters. **c** Time courses of gas evolution during photocatalytic Z-scheme OWS under simulated sunlight using Cr₂O₃/Pt (MW_EG)/Ir (MW_H2O)-modified SrTaO₂N nanocrystals (solid circles) or large-sized SrTaO₂N (precursor: Ta₂O₅/SrCO₃/SrCl₂, Supplementary Fig. 13) (open squares) as the HEP, Ir-CoFeO_x/BiVO₄ as the OEP, and [Fe(CN)₆]³⁻/[Fe(CN)₆]⁴⁻ as a redox mediator. Conditions: HEP, 50 mg; Ir-CoFeO_x/BiVO₄, 100 mg; 150 mL of 25 mM phosphate buffer solution (pH = 6) containing K₄[Fe(CN)₆] (5 mM); light source, solar simulator (AM 1.5 G, 0.87 sun); irradiation area for solar simulator, 9.3 cm²; background pressure, 5 kPa.

a representative H₂-evolution cocatalyst (Fig. 3a). Specifically, the addition of Ir and Cr₂O₃ increased the H₂-evolution rate approximately twofold (photocatalyst vi versus v in Fig. 3a) and sixfold (photocatalyst iii versus v in Fig. 3a), respectively, compared with the case when either of them were absent. Because individual Ir- or Cr₂O₃-modified SrTaO₂N exhibited negligible activities toward H₂ evolution (Supplementary Fig. 19), the promotion effect of both components most likely resulted from their interactions with Pt that promote charge separation and transfer and/or promote surface reactions. This result motivated us, above all, to investigate the nanostructure of the Cr₂O₃/Pt (MW_EG)/Ir (MW_H2O) cocatalyst and, in particular, the interactions between different cocatalyst components.

Decoration with Ir (MW_H2O) formed highly dispersed IrO₂ species according to X-ray photoelectron spectroscopy (XPS) (Supplementary Fig. 20a) and ADF STEM (Supplementary Fig. 21a) analyses. Further decoration with Pt (MW_EG) via a microwave heating process with EG as a reducing agent generated noticeable tiny nanoparticles on the surface of the SrTaO₂N (Supplementary Fig. 21b). This step not only produced metallic Pt (Supplementary Fig. 22a) but also reduced approximately one-half of the previous IrO₂ decoration species to metallic Ir (Supplementary Fig. 20c). This result is consistent with the observation that, when Ir (MW_H2O)/SrTaO₂N was subjected to a similar EG-mediated microwave treatment (denoted as Ir (MW_H2O-MW_EG)/SrTaO₂N), approximately 43% of the IrO₂ species was reduced to metallic Ir (Supplementary Fig. 20b). Interestingly, the XPS 4f peak

position for Ir⁰ in Pt (MW_EG)/Ir (MW_H2O)/SrTaO₂N (Supplementary Fig. 20c) was negatively shifted by approximately 0.8 eV compared with that for Ir⁰ in the Ir (MW_H2O-MW_EG)/SrTaO₂N specimen (Supplementary Fig. 20b). Meanwhile, the XPS 4f peak position for Pt⁰ in Pt (MW_EG)/Ir (MW_H2O)/SrTaO₂N (Supplementary Fig. 22b) was positively shifted by approximately 0.6 eV compared with that for Pt⁰ in the Pt (MW_EG)/SrTaO₂N specimen (Supplementary Fig. 22a). These shifts clearly point to strong electronic interactions between Ir⁰ and Pt⁰ species in Pt (MW_EG)/Ir (MW_H2O)/SrTaO₂N, which indicates the formation of metal alloys[17,22,23]. STEM–EDS mapping and line-scan images (Fig. 3b–e) further indicate that the fed Ir and Pt species in Pt(MW_EG)/Ir (MW_H2O)/SrTaO₂N dominantly form Ir–Pt alloy nanoparticles because both the Ir and Pt elements were confined across the area of individual nanoparticles. Cr₂O₃ was lastly photodeposited, forming Ir–Pt alloy@Cr₂O₃ core–shell nanostructured particles in Cr₂O₃/Pt (MW_EG)/Ir (MW_H2O)/SrTaO₂N, as is evident in the STEM–EDS elemental maps (Fig. 3f–i) and HRTEM image (Fig. 3j). Notably, on the basis of the identification of both Cr₂O₃ and Cr(OH)₃ species by XPS (Supplementary Fig. 23) and previous studies[6,24], the Cr₂O₃ shell was composed of amorphous Cr(III)O₁.₅₋ₘ(OH)₂ₘ·xH₂O and the Cr₂O₃ decoration did not alter the chemical state of Ir (Supplementary Fig. 20d) or Pt (Supplementary Fig. 22c).

The transient absorption (TA) kinetic profiles probed at 2000 nm (Fig. 3k) reflect the intraband transition of long-lived electrons in oxynitride materials[15,25]. The observed decrease in the TA signal

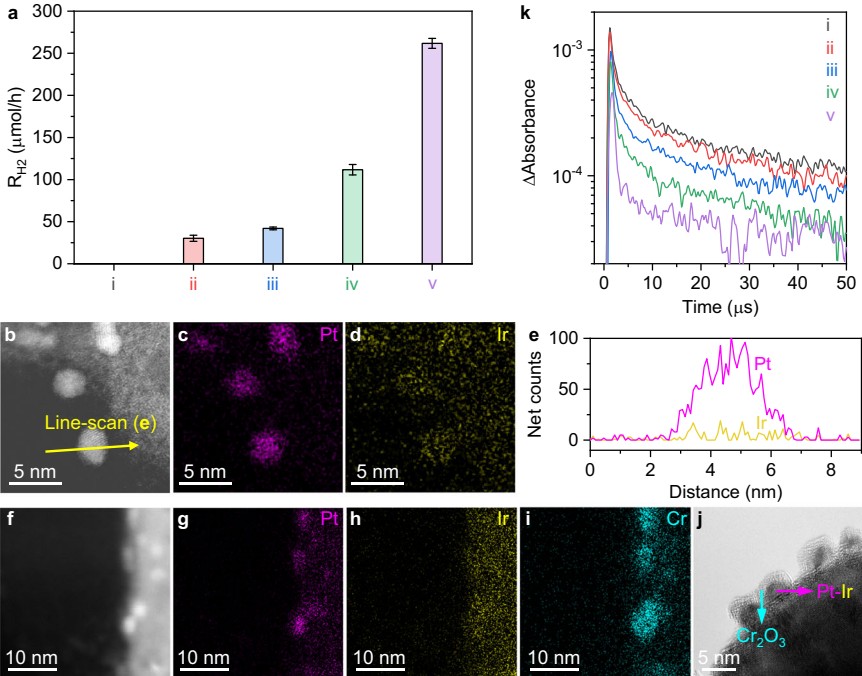

**Fig. 3 | Nanostructure and characteristics of Cr₂O₃/Pt (MW_EG)/Ir (MW_H2O) cocatalyst promoting H₂ evolution. a** Photocatalytic H₂-evolution rate (calculated in the first one hour) for bare and different cocatalyst-modified SrTaO₂N nanocrystals: i, bare SrTaO₂N; ii, Pt (MW_EG)/SrTaO₂N; iii, Pt (MW_EG)/Ir (MW_H2O)/SrTaO₂N; iv, Cr₂O₃/Pt (MW_EG)/SrTaO₂N; v, Cr₂O₃/Pt (MW_EG)/Ir (MW_H2O)/SrTaO₂N. Error bars indicate the standard deviation of three measurements. ADF STEM image (**b**), and Pt (**c**) and Ir (**d**) EDS mapping images of a cross-sectional Pt (MW_EG)/Ir (MW_H2O)/SrTaO₂N sample. **e** Pt and Ir EDS line-scan spectra across the nanoparticle marked with a yellow arrow in (**b**). ADF STEM image (**f**), and Pt (**g**), Ir (**h**), and Cr (**i**) EDS mapping images of a cross-sectional Cr₂O₃/Pt (MW_EG)/Ir (MW_H2O)/SrTaO₂N sample. **j** HRTEM image of the surface of a Cr₂O₃/Pt (MW_EG)/Ir (MW_H2O)-modified SrTaO₂N nanocrystal. **k** TA kinetic profiles of photoexcited electrons probed at 2000 nm over bare and different cocatalyst-modified SrTaO₂N nanocrystals in subfigure (**a**).

intensity upon decoration with a cocatalyst indicates electron transfer from the SrTaO₂N to the cocatalyst for the water reduction reaction. The slight decrease (from line i to ii in Fig. 3k) in the TA signal intensity upon decoration with Pt (MW_EG) indicates that electrons were not effectively injected into the Pt. Decoration with Pt (MW_EG)/Ir (MW_H2O) resulted in a more substantial decrease (from line i to iii in Fig. 3k) in the TA signal intensity. This result confirms that the Ir−Pt alloy acts as a more efficient electron collector and H₂-evolution catalyst than Pt alone, consistent with the higher H₂-evolution rate for Pt(MW_EG)/Ir (MW_H2O)/SrTaO₂N compared with that for Pt(MW_EG)/SrTaO₂N (Fig. 3a). The most notable decrease in the TA signal intensity was observed when Pt or Ir−Pt alloy was capped with a Cr₂O₃ overlayer (from line ii to iv and from iii to v in Fig. 3k). This result provides direct evidence that Cr₂O₃ greatly promotes electron transfer, consistent with our recent finding that a Cr₂O₃ coating on Rh nanoparticles on a fluorine-doped tin oxide (FTO) substrate substantially enhanced the current density[26]. The enhancement was most likely attributable to the Cr₂O₃ overlayer inhibiting hole transfer from the vicinity of the Pt or Ir−Pt alloy nanoparticles and thereby greatly reducing charge recombination at the metal nanoparticles or the SrTaO₂N−metal interfaces. Moreover, the Cr₂O₃ shell is known to function as a molecular sieve to allow the permeation of H⁺, H₂, and H₂O species while preventing the oxidized species generated by holes from reaching the metal core and being reduced back[24,26,27]. Therefore, the deposition of Cr₂O₃ improved the H₂-evolution rate approximately sixfold (Fig. 3a) and notably improved the photocatalytic durability by suppressing unfavorable side reactions (Supplementary Fig. 24).

## O₂-evolution performance of CoO_x-modified SrTaO₂N nanocrystals

Upon modification with a classic cobalt oxide (CoO_x) cocatalyst (Supplementary Fig. 25) with an optimized feed concentration of 1.0 wt% (relative to the mass of SrTaO₂N; Supplementary Fig. 26), the SrTaO₂N nanocrystals also exhibited high photocatalytic O₂-evolution performance from an aqueous AgNO₃ solution (Fig. 4a). The associated AQYs were estimated to be 9.0%, 6.6%, and 0.2% at 422, 479, and 580 nm, respectively (Fig. 4b). These values are higher than most of those reported for SrTaO₂N-based OEPs with CoO_x as the cocatalyst (Supplementary Table 2). This result demonstrates again the importance of the developed approach to afford high-quality SrTaO₂N single nanocrystals. Consistent with the findings of previous studies[16], CoO_x is critical for achieving efficient O₂ evolution (Fig. 4c) because it can efficiently capture holes for the water oxidation reaction. Interestingly, the TA signal intensity probed at 5000 nm (Fig. 4d), which reflects the dynamics of free and/or shallowly trapped electrons[28], was enhanced by approximately one order of magnitude following decoration by CoO_x. This suggests that one-step-excitation OWS over this material will be a viable process in the near future if these populous long-lived electrons resulting from the CoO_x modification can be further extracted and used for H₂ evolution via appropriate surface modification and/or cocatalyst design.

This work demonstrates that highly crystalline single nanocrystals of perovskite-type tantalum oxynitrides can be easily synthesized by thermal nitridation of mixtures of TaS₂, metal hydroxides, and molten salts. Upon modification with a tailored Ir−Pt alloy@Cr₂O₃ cocatalyst, the SrTaO₂N nanocrystals produced by this approach evolved H₂ around two orders of magnitude more efficiently than the previously reported SrTaO₂N photocatalysts, with an apparent quantum yield of 3% at the wavelength of 420 nm, from a methanol aqueous solution, and an STH energy conversion efficiency of 0.15% in Z-scheme water splitting. The CoO_x-modified SrTaO₂N nanocrystals also evolved oxygen efficiently, surpassing most of the reported SrTaO₂N photocatalysts. We envision that appropriate surface and cocatalyst modifications would enable efficient one-step-excitation OWS over these ATaO₂N (A = Sr, Ca,

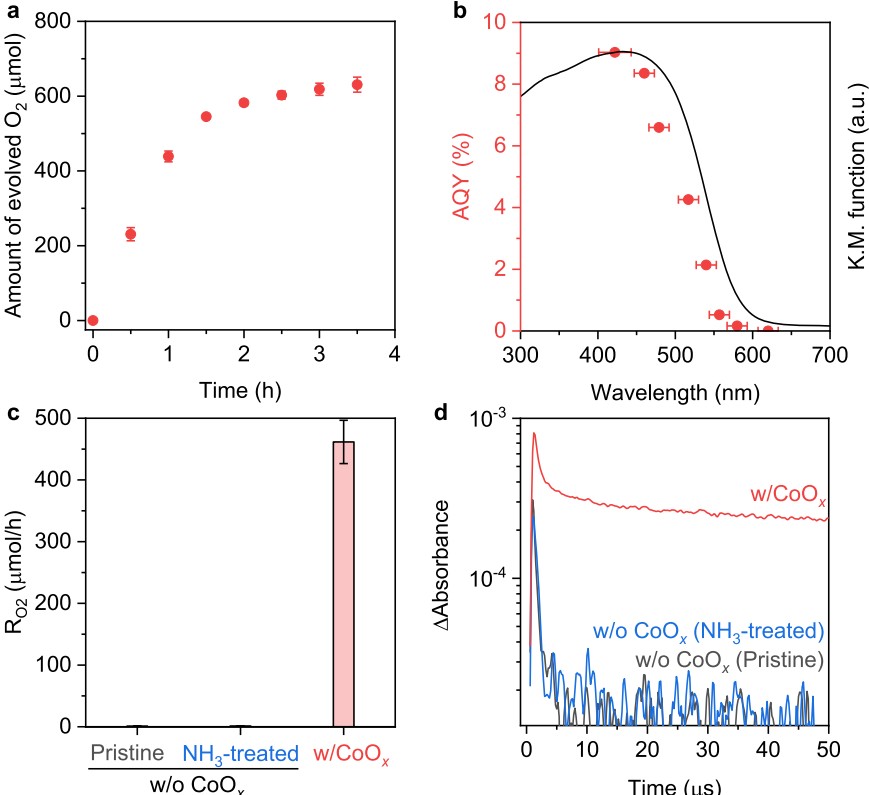

**Fig. 4 | O$_2$-evolution performance of SrTaO$_2$N nanocrystal photocatalyst.** Time course (**a**) of photocatalytic O$_2$-evolution reaction in 20 mM AgNO$_3$ solution over a CoO$_x$-modified SrTaO$_2$N photocatalyst under visible-light irradiation and the associated AQY at various wavelengths (**b**). The above reactions were carried out under illumination from a Xe lamp (300 W, $\lambda \geq 420$ nm) with or without various bandpass filters. The solid line in (**b**) indicates the UV–vis diffuse-reflectance spectrum of SrTaO$_2$N. Photocatalytic O$_2$-evolution rates calculated at 0.5 h (**c**) and TA kinetic profiles of photoexcited electrons probed at 5000 nm (**d**) over pristine and NH$_3$-treated SrTaO$_2$N without CoO$_x$ and over CoO$_x$/SrTaO$_2$N. Error bars in (**a**, **c**) indicate the standard deviation of three measurements.

Ba) nanocrystals in the near future. The proposed synthetic strategy is expected to be applicable to a broad range of perovskite-type transition-metal oxynitride single nanocrystals, and can advance the manufacturing of long-wavelength-responsive particulate photocatalysts for efficient solar energy conversion.

## Methods

### Synthesis of nanocrystalline $A$TaO$_2$N ($A$ = Sr, Ca, Ba)

$A$TaO$_2$N ($A$ = Sr, Ca, Ba) nanocrystals were synthesized by heating of a mixture of TaS$_2$, $A$(OH)$_2$, and SrCl$_2$ with a molar ratio of 1:2.5:1 under gaseous NH$_3$ flow. In a typical synthesis of SrTaO$_2$N nanocrystals, 1.20 g of TaS$_2$ (99%; Kojundo Chemical Laboratory, Supplementary Fig. 1), 3.25 g of Sr(OH)$_2$·8H$_2$O (90.0+%; FUJIFILM Wako Pure Chemical), 0.78 g of SrCl$_2$ (98.0%; Kanto Chemical), and 5 mL of ethanol were well mixed in an agate mortar with the aid of sonication and agitation. After desiccation by a mild heating process, the resultant powder was loaded into an alumina crucible and heated at 1223 K for 3 h under a 200 mL min$^{-1}$ flow of gaseous NH$_3$. SrTaO$_2$N nanocrystals were obtained after the resultant solids were rinsed and then dried overnight at 313 K under vacuum. CaTaO$_2$N and BaTaO$_2$N nanocrystals were synthesized by a similar procedure in which Ca(OH)$_2$ (99.9%; FUJIFILM Wako Pure Chemical) and Ba(OH)$_2$·8H$_2$O (98.0%; FUJIFILM Wako Pure Chemical) were used, respectively, instead of Sr(OH)$_2$·8H$_2$O.

### Cocatalyst modification for SrTaO$_2$N

For H$_2$-evolution reactions, SrTaO$_2$N was decorated with Ir and Pt sequentially by microwave heating in water and EG, respectively, and finally with Cr$_2$O$_3$ by photodeposition via a modified version of a recently reported method[29]. The resultant material is denoted herein as Cr$_2$O$_3$/Pt (MW$_{EG}$)/Ir (MW$_{H2O}$)/SrTaO$_2$N. In a typical process, SrTaO$_2$N nanoparticles were first dispersed in distilled water (15 mL) containing the desired amount of IrCl$_3$·3H$_2$O (99.9%; Kanto Chemical) and the suspension was subsequently heated at 423 K for 10 min using a microwave reactor (Anton Paar, Monowave 200) while the suspension was stirred at 1000 rpm. After the sample cooled naturally, it was washed, filtered, and then dried at 313 K under vacuum to obtain Ir (MW$_{H2O}$)/SrTaO$_2$N. This sample was further subjected to similar microwave heating in a mixture of 13 mL of EG and 2 mL of distilled water containing the required amount of H$_2$PtCl$_6$·6H$_2$O (>98.5%; Kanto Chemical) under the same conditions, resulting in Pt (MW$_{EG}$)/Ir (MW$_{H2O}$)/SrTaO$_2$N. After washing and filtration, the wet Pt (MW$_{EG}$)/Ir (MW$_{H2O}$)/SrTaO$_2$N was dispersed in an aqueous methanol solution (13 vol%) containing the required amount of K$_2$CrO$_4$ (99.0%; Kanto Chemical). After complete degassing, the suspension was irradiated with visible light ($\lambda \geq 420$ nm) for 0.5 h; Cr$_2$O$_3$/Pt (MW$_{EG}$)/Ir (MW$_{H2O}$)-modified SrTaO$_2$N was obtained.

For O$_2$-evolution reactions, SrTaO$_2$N was modified with the CoO$_x$ cocatalyst by impregnation, followed by heating under a gaseous NH$_3$ flow. A quantity of the SrTaO$_2$N powder was immersed in an aqueous solution containing the required amount of Co(NO$_3$)$_2$·6H$_2$O (>98.0%; Kanto Chemical) as a Co precursor, and the resultant slurry was continuously stirred with intense sonication for 2 min to disperse the SrTaO$_2$N. After the slurry was dried on a hot-water bath, the powder sample was heated at 1173 K for 1 h under a 200 mL min$^{-1}$ flow of gaseous NH$_3$ to obtain the CoO$_x$/SrTaO$_2$N nanoparticulate photocatalyst.

### Preparation of Ir-CoFeO$_x$/BiVO$_4$

Ir-CoFeO$_x$/BiVO$_4$ was prepared according to the method reported elsewhere[20]. NH$_4$VO$_3$ (10 mmol; 99.0%; FUJIFILM Wako Pure Chemical)

and $Bi(NO_3)_3 \cdot 5H_2O$ (10 mmol; 99.5%; FUJIFILM Wako Pure Chemical) were dissolved in 2.0 M nitric acid solution, whose pH was then adjusted to ~0.5 by addition of an $NH_3$ solution (25–28 wt%). The mixed solution was strongly stirred until a light-yellow precipitate was observed; the precipitate was further aged for ~2 h and then transferred to a Teflon-lined stainless-steel autoclave for a 24 h hydrothermal treatment at 473 K. $BiVO_4$ was obtained after the collected powder was washed with distilled water and dried in vacuum at 313 K for 6 h. The $BiVO_4$ was further suspended in 150 mL of 25 mM phosphate buffer solution (pH = 6) containing a calculated amount of $Na_2IrCl_6$ (0.8 wt% Ir, relative to the mass of $BiVO_4$), $Co(NO_3)_2$ (0.2 wt% Co, relative to the mass of $BiVO_4$), and $K_3[Fe(CN)_6]$ (0.1 mM). After irradiation with UV–vis light for 2 h under static air conditions, Ir-CoFeO$_x$/BiVO_4 was collected, washed, and dried for further use.

## Characterization of materials

The crystal phases were characterized by XRD analysis using a Rigaku MiniFlex 300 powder diffractometer equipped with a Cu Kα radiation source ($\lambda = 1.5418$ Å). Diffuse-reflectance spectra were acquired with an ultraviolet–visible–near-infrared spectrometer (V-670, JASCO) and converted from reflectance into the Kubelka–Munk function. The contents of Sr and Ta metals in the $SrTaO_2N$ were determined by inductively coupled plasma atomic emission spectroscopy (ICP-AES; ICPS-8100, Shimadzu). The oxygen, and nitrogen contents of the $SrTaO_2N$ were determined with an oxygen–nitrogen analyzer (Horiba, EMGA-920); while the sulfur content was determined with a carbon-sulfur analyzer (Horiba, EMIA-Pro). XPS analysis was carried out using a PHI Quantera II spectrometer equipped with an Al Kα radiation source. All binding energies were referenced to the C 1$s$ peak (284.8 eV) arising from adventitious carbon. Scanning electron microscopy (SEM) and TEM images were acquired with a Hitachi HD-2300A scanning transmission electron microscope using the SEM and TEM modes, respectively. STEM images, EDS mapping images and SAED patterns were recorded using a JEOL JEM-ARM200F Cold FE (Cs-STEM), a JEOL JEM-ARM200F Thermal FE (Cs-STEM) and a JEOL JEM-2800 equipped with an Oxford Instruments X-MAX 100TLE SDD detector, respectively. The cross-sectional sample for STEM observation was made by Ar ion milling using a JEOL EM-09100IS ion slicer.

Mid-infrared (IR) time-resolved TA spectroscopic investigations were performed using a pump–probe nanosecond system equipped with a Nd:YAG laser (Continuum, Surelite I; duration: 6 ns) and custom-built spectrometers[17,28,30]. Photoexcited charge carriers were monitored in the mid-IR region, with probe energies from 6000 to 1200 cm$^{-1}$ (from 1667 to 8333 nm), providing dynamics of photogenerated electrons in the photocatalysts[17,28,30]. The IR probe light from a $MoSi_2$ coil was focused on the photoexcited sample, and the transmitted IR beam was then passed through the monochromatic grating spectrometer. The transmitted light was subsequently detected by a mercury-cadmium-telluride detector (Kolmar). To excite the photocarriers in $SrTaO_2N$ with and without cocatalysts, 440 nm laser pulses with a fluence of 500 μJ pulse$^{-1}$ generated from an optical parametric oscillator were used. The output electrical signal was amplified using an alternating-current coupled amplifier (Stanford Research Systems (SR560), bandwidth: 1 MHz). The time resolution of the spectrometer was limited to 1 μs by the bandwidth of the amplifier. For data acquisition, 1000 TA signals were accumulated to generate a decay profile at the probe wavelength. For preparation of the sample film, a suitable amount of bare or cocatalyst-loaded $SrTaO_2N$ was dispersed on water and then drop-cast onto a $CaF_2$ substrate to obtain a film with a density of 1.09 mg cm$^{-2}$; the film was then dried in air overnight. TA spectroscopic measurements were carried out under $N_2$ ambient (20 Torr) and at room temperature.

## Photocatalytic reactions of $H_2$ and $O_2$ evolution and Z-scheme OWS

All photocatalytic reactions were carried out at 288 K under a background pressure of 5 kPa in a Pyrex top-illuminated reaction vessel connected to a closed gas circulation system. For the photocatalytic $H_2$-evolution reaction, 150 mg of photocatalyst was well dispersed in 150 mL of an aqueous methanol solution (13 vol%) without adjustment of the pH. After air was completely removed from the reaction slurry by evacuation, Ar gas was introduced to generate a background pressure of approximately 5 kPa and the reactant solution was irradiated with a 300 W Xe lamp equipped with a cold mirror and a cut-off filter (L42, $\lambda \geq 420$ nm). For the photocatalytic $O_2$-evolution reaction, 150 mg of photocatalyst together with 100 mg of $La_2O_3$ as a pH buffer was well dispersed in 150 mL of an aqueous 20 mM $AgNO_3$ solution. After air was completely removed from the reaction vessel, the photocatalyst suspension was irradiated with a 300 W Xe lamp equipped with a cold mirror and a cut-off filter (L42, $\lambda \geq 420$ nm). For the photocatalytic Z-scheme OWS reaction, $Cr_2O_3$/Pt (MW$_{EG}$)/Ir (MW$_{H_2O}$)-modified $SrTaO_2N$ (50 mg) as the HEP and Ir-FeCoO$_x$-modified $BiVO_4$ (100 mg) as the OEP were dispersed in 150 mL of 25 mM sodium phosphate buffer solution (pH = 6.0) containing $K_4[Fe(CN)_6]$ (5 mM). After air was completely removed from the reaction slurry by evacuation, Ar gas was introduced to generate a background pressure of approximately 5 kPa and the suspension was irradiated by a solar simulator (SAN-EI Electronic, XES40S1, AM 1.5 G, 87 mW cm$^{-2}$). The top window of the reaction vessel was covered with a mask to confine the irradiated sample area to 9.3 cm$^2$. The gaseous products evolved during these reactions were analyzed using an integrated online gas chromatography system consisting of a GC-8A chromatograph (Shimadzu) equipped with molecular sieve 5 Å columns and a thermal conductivity detector, with Ar as the carrier gas.

## AQY measurements

The AQY for the photocatalytic reactions was calculated according to the equation

$$AQY(\%) = [A \times R]/I \times 100,$$

where $R$ and $I$ are the rate of gas evolution and the incident photon flux, respectively, and $A$ is the number of electrons needed to generate one molecule of $H_2$ or $O_2$ (i.e., two or four for photocatalytic sacrificial $H_2$ or $O_2$ evolution, respectively, and four for $H_2$ evolution in Z-scheme water splitting based on two-step photoexcitation). The photocatalytic reactions were carried out using the same experimental setup and conditions described above, except for the use of bandpass filters with central wavelengths of 422, 460, 479, 517, 540, 557, 580, and 620 nm. The FWHM of the 422 nm bandpass filter was 21 nm, whereas that of the other bandpass filters was 13 nm. The number of incident photons was measured using an LS-100 grating spectroradiometer (EKO Instruments).

## STH energy conversion efficiency measurements

The water-splitting reaction was performed under simulated solar radiation generated with a solar simulator (SAN-EI Electronic, XES40S1, AM 1.5 G). The STH conversion efficiency was calculated as

$$STH(\%) = (R_{H_2} \times \Delta G)/(P \times S) \times 100,$$

where $R_{H_2}$, $\Delta G$, $P$, and $S$ denote the rate of $H_2$ evolution during the OWS reaction, the Gibbs energy for the OWS reaction (237 kJ mol$^{-1}$ at 288 K), the energy intensity (87 mW cm$^{-2}$) of the AM 1.5 G solar radiation used (equivalent to 0.87 sun), and the irradiated sample area (9.32 cm$^2$), respectively.

## Data availability

The authors declare that the data supporting the findings of this study are available within the paper and its Supplementary Information files. Source data are provided with this paper.

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

## Acknowledgements

This work was financially supported by the Artificial Photosynthesis Project of the New Energy and Industrial Technology Development Organization (NEDO), and the Advanced Research Infrastructure for Materials and Nanotechnology (ARIM) of the Ministry of Education, Culture, Sports, Science and Technology (MEXT), Japan (grant nos. JPMXP1222UT0023 and JPMXP1223UT0004). J.X. thanks Ms. Michiko Obata (Shinshu University) and Ms. Keiko Kato (University of Tokyo) for their assistance in performing XPS and elemental analyses, respectively. W. L. thanks 2021 MEXT Scholarship with Embassy Recommendation and China Scholarship Council Scholarship (grant no. 202008440289) for the financial support of his doctoral study in Japan.

## Author contributions

J.X. and K.D. conceived and designed the research. J.X. prepared the photocatalyst materials, performed conventional material characterization and photocatalytic performance evaluation. M.N. and N.S. carried out STEM–EDS-related measurements and analyzed the resulting data. J.J.M.V. and A.Y. conducted the TAS experiments and analyzed the resulting data. W.L., K.C., and X.T. helped with the preparation of $BiVO_4$ and establishment of the Z-scheme OWS system. J.X., T.H., M.N., J.J.M.V., T.T., Y.I., and K.D. discussed the results. K.D. supervised the entire research work. J.X., T.H., and K.D. wrote and revised the paper with contributions from the other authors.

## Competing interests

J.X., T.H. and K.D. of Shinshu University and Y.I. of Japan Technological Research Association of Artificial Photosynthetic Chemical Process hold

a patent related to this work (Japanese Unexamined Patent Application Publication No. 2023-031163). The remaining authors declare no competing interests.

## Additional information

**Peer review information** : *Nature Communications* thanks Gang Liu and the other, anonymous, reviewer(s) for their contribution to the peer review of this work. A peer review file is available.

