## [Peer review file · Nature Communications]

Sub-50 nm Perovskite-Type Tantalum-Based Oxynitride Single Crystals with Enhanced Photoactivity for Water SplittingEditorial Note: This manuscript has been previously reviewed at another journal that is not operating a transparent peer review scheme. This document only contains reviewer comments and rebuttal letters for versions considered at Nature Communications.

REVIEWER COMMENTS

Reviewer #1 (Remarks to the Author):

The authors have adequately addressed my concerns. I support its publication in Nature Communications without further reservations.

Reviewer #2 (Remarks to the Author):

In this work named "Sub-50 nm Perovskite-Type Tantalum-Based Oxynitride Single Crystals with Enhanced Photoactivity for Water Splitting", the authors prepared a series of ATaO₂N single nanocrystal with high crystallinity and minimizing particle size using a modified flux method. Among them, SrTaO₂N shows a recorded photocatalytic hydrogen evolution activity by modifying Ir–Pt alloy@Cr₂O₃ cocatalyst. The constructed Z-scheme system exhibits an STH energy conversion efficiency of 0.15%. This work could make an important construction in designing the perovskite-type tantalum oxynitrides with high solar to hydrogen conversion efficiency. I would like to recommend it for publication after appropriate revisions. The detailed comments are as follows.

1. In another work (J. Am. Chem. Soc. 2023, 145, 7, 3839–3843) from the Domen laboratory, the authors reported a synthesis of about 120 nm SrTaO₂N for photocatalytic water splitting. the prepared samples show a one-step-excitation overall water splitting activity. The reported SrTaO₂N should be discussed and used as a reference sample to display the importance and novelty of this work.
2. Although the authors have successfully synthesized ATaO₂N single nanocrystals with high crystallinity and minimizing particle size. The possible mechanism of the method is still unclear, which will weaken the universality of this method and make it difficult to provide theoretical guidance for the preparation of other similar compounds.
3. Following question 2, is it possible to prepare the SrTaO₂N with different particle sizes to study the effect of the particle size on photocatalytic water splitting?
4. As reported, by using SrCl₂ as the molten salt, the alumina crucible could be used as the aluminum source for element doping. In this work, what is the amount of Al in the prepared SrTaO₂N and the effect of doped Al for photocatalytic water splitting activity?
5. The authors attribute the high photocatalytic water splitting activity of the prepared SrTaO₂N to overcoming the long distances for the photoexcited charge carriers to migrate to reach active sites on the surface of common ATaO₂N. To confirm this conclusion, the photophysical properties of samples should be discussed by using some proper methods such as fluorescence spectroscopy, surface photovoltage intensity, and Transient absorption spectroscopy.
6. How about the one-step-excitation overall water splitting activity of the prepared SrTaO₂N?

Reviewer #3 (Remarks to the Author):

The authors have satisfactorily replied to my concern and I believe the manuscript is suitable for Nature Communications.

Response to The Reviewers' Comments

We were very pleased to read the three referee reports on our manuscript. We have now addressed the different advices and concerns raised by Reviewer 2 and made a revised version of the article based on the additions/changes implemented. In what follows, we will address each of the comments, including the actions taken to revise our manuscript. The files highlighting the changes made (in yellow) in the new manuscript and Supplementary Information are also provided for the review process.

Reviewer #1

Comment. The authors have adequately addressed my concerns. I support its publication in *Nature Communications* without further revisions.

Response: We sincerely thank the reviewer for his/her previous constructive comments/suggestions for improving the quality of our work.

Reviewer #2

General Comment. In this work named “Sub-50 nm Perovskite-Type Tantalum-Based Oxynitride Single Crystals with Enhanced Photoactivity for Water Splitting”, the authors prepared a series of $ATaO_2N$ single nanocrystal with high crystallinity and minimizing particle size using a modified flux method. Among them, $SrTaO_2N$ shows a recorded photocatalytic hydrogen evolution activity by modifying Ir–Pt alloy@ Cr_2O_3 cocatalyst. The constructed Z-scheme system exhibits an STH energy conversion efficiency of 0.15%. This work could make an important construction in designing the perovskite-type tantalum oxynitrides with high solar to hydrogen conversion efficiency. I would like to recommend it for publication after appropriate revisions.

Response: We sincerely thank the referee for the positive comments and hope

to improve the quality of the present work following the referee's constructive comments and suggestions.

Comment 1. In another work (J. Am. Chem. Soc. 2023, 145, 7, 3839–3843) from the Domen laboratory, the authors reported a synthesis of about 120 nm SrTaO₂N for photocatalytic water splitting. The prepared samples show a one-step-excitation overall water splitting activity. The reported SrTaO₂N should be discussed and used as a reference sample to display the importance and novelty of this work.

Response: We sincerely thank the reviewer for this comment. We note that the previously developed SrTaO₂N [J. Am. Chem. Soc. **2023** 145 (7), 3839-3843] prepared from a Ta₂O₅/NaOH/SrCl₂ precursor inevitably contained a pinch of Ta₃N₅ as a byproduct, and the particle-size distribution (70–250 nm) was relatively wide. In comparison, the developed approach in this work can afford highly crystalline single nanocrystals of SrTaO₂N without forming any impurities and the particle sizes were narrowly controlled in the range of 25–85 nm (below 100 nm and averagely 52 nm, Figure 1d). It provides, to the best of knowledge, the first method that can prepared perovskite-type transition-metal oxynitride single crystals simultaneously possessing a high degree of crystallinity, a particle size of several tens of nanometers and a low defect density; all these properties are actively pursued for developing overall-water-splitting (OWS) photocatalysts [Acc. Chem. Res. **2023** 56 (7), 878-888]. Moreover, the developed SrTaO₂N nanocrystals exhibited around 2.4 times higher photocatalytic H₂ evolution activity than the previous SrTaO₂N (Ta₂O₅/NaOH/SrCl₂) specimen (Figure R1). Although one-step-excitation OWS was not successful in the present SrTaO₂N nanocrystals when depositing the same cocatalyst reported [J. Am. Chem. Soc. **2023** 145 (7), 3839-3843], these tantalum-based oxynitride nanocrystals would have great potential to split water efficiently via single-step excitation once specific surface/cocatalyst

modifications over these materials can be well implemented.

Figure R1. (a) Time courses for photocatalytic H₂ evolution and (b) the associated H₂-evolution rates over the previously developed SrTaO₂N (prepared from a Ta₂O₅/NaOH/SrCl₂ precursor¹) and single-nanocrystal SrTaO₂N (prepared from a TaS₂/Sr(OH)₂/SrCl₂ precursor) modified with the same Cr₂O₃/Pt (MW_{EG})/Ir (MW_{H₂O}) (Cr, 0.5 wt%; Pt, 1 wt%; Ir, 0.5 wt%) cocatalysts in an aqueous methanol solution under visible light ($\lambda \geq 420$ nm).

Action: We add Figure R1 as Supplementary Fig. 16 and cite the paper mentioned [*J. Am. Chem. Soc.* **2023** 145 (7), 3839-3843] as Supplementary Reference 1 in the revised Supplementary Information.

Below Supplementary Fig. 16, we add a note “*Note: The reference SrTaO₂N (Ta₂O₅/NaOH/SrCl₂) specimen was prepared according to our previously reported method (i.e., thermal nitridation of a Ta₂O₅/NaOH/SrCl₂ precursor with a molar ratio of 1:1:4 under 200 mL/min NH₃ at 1223 K for 5 h)¹, which contained a pinch of Ta₃N₅ as a byproduct and exhibited a relatively wide particle-size distribution (70–250 nm). This reference sample (R_{H₂} = 107 μmol/h) exhibited 2.4 times lower performance for photocatalytic H₂ evolution compared with the single-nanocrystal SrTaO₂N (R_{H₂} = 262 μmol/h) developed in this work.*”

In addition, the original sentence “*Moreover, the single-nanocrystal SrTaO₂N evolved H₂ four times higher than the polycrystalline SrTaO₂N exhibiting*

aggregates composed of several polycrystalline nanoparticles with an average size of 63 nm (see detailed discussion in Supplementary Fig. 14).” in the Section “H₂-evolution and Z-scheme water-splitting performance of the SrTaO₂N-nanocrystal photocatalyst” of the main text

was modified to

“Moreover, the single-nanocrystal SrTaO₂N evolved H₂ four times higher than the polycrystalline SrTaO₂N exhibiting aggregates composed of several polycrystalline nanoparticles with an average size of 63 nm (see detailed discussion in Supplementary Fig. 15) and 2.4 times higher than the SrTaO₂N previously developed from a Ta₂O₅/NaOH/SrCl₂ precursor¹⁷ (Supplementary Fig. 16).”

Comment 2. Although the authors have successfully synthesized ATaO₂N single nanocrystals with high crystallinity and minimizing particle size. The possible mechanism of the method is still unclear, which will weaken the universality of this method and make it difficult to provide theoretical guidance for the preparation of other similar compounds.

Response: We thank the reviewer for this constructive comment. To address the issue, additional experiments (Figure R2) were performed. As shown in Fig. R2a and b, direct nitridation of TaS₂ under NH₃ generated Ta₃N₅ retaining the several-micron-sized stacked-layer structure of TaS₂ (Supplementary Fig. 1a). In contrast, nitridation of TaS₂ in the presence of SrCl₂ for 0.5 h formed aggregates dominantly composed of monodispersed Ta₃N₅ nanoparticles with an average size of 29 nm (Figure R2c-e). The generation of minor SrTaO₂N in the latter case is due to the favorable water uptake by SrCl₂ in the precursor. This difference indicates that the SrCl₂ molten-salt could promote the decomposition of TaS₂ into nanoscale fragments during the thermal nitridation process, which is presumably a key reason for the formation of ATaO₂N single

nanocrystals upon thermal nitridation of a mixture of $\text{TaS}_2/\text{Sr}(\text{OH})_2/\text{SrCl}_2$. Nevertheless, explicit elucidation of the detailed compound formation mechanism would require further intensive study through *in-situ* spectroscopy and microscopy, which may be considered in a specialized work in the near future.

On the other hand, we note again that the developed approach can afford perovskite-type tantalum-based oxynitride single crystals simultaneously possessing a high degree of crystallinity, a particle size of several tens of nanometers and a low defect density; and the light-absorption edge wavelengths (Fig. 1j) and degrees of crystallinity (Supplementary Fig. 9) can be readily manipulated by modifying the synthetic parameters (such as the alkaline-earth metal and molten salt in the precursor). This versatility of the synthesis method for unique functional materials will justify the lack of comprehensive understanding in the particle/compound formation mechanism.

Figure R2. (a) SEM image and (b) XRD pattern of the material generated by nitridation of TaS_2 at 1223 K for 3 h. (c) SEM image, (d) particle-size distribution and (e) XRD pattern of the material generated by nitridation of $\text{TaS}_2/\text{SrCl}_2$ with

a molar ratio of 1:1 at 1223 K for 0.5 h. The mean value and standard deviation (SD) of the particle sizes in subfigure d were determined by Gaussian fitting (red line).

Action: We add Figure R2 as Supplementary Fig. 9 with a note

“Note: As shown in Supplementary Fig. 9a and b, direct nitridation of TaS₂ under NH₃ generated Ta₃N₅ retaining the several-micron-sized stacked-layer structure of TaS₂ (Supplementary Fig. 1a). In contrast, nitridation of TaS₂ in the presence of SrCl₂ for 0.5 h formed aggregates dominantly composed of monodispersed Ta₃N₅ nanoparticles with an average size of 29 nm (Supplementary Fig. 9c-e). The generation of minor SrTaO₂N in the latter case is due to the favorable water uptake by SrCl₂ in the precursor. This difference indicates that the SrCl₂ molten-salt could promote the decomposition of TaS₂ into nanoscale fragments during the thermal nitridation process, which is presumably a key reason for the formation of ATaO₂N single nanocrystals upon thermal nitridation of a mixture of TaS₂/Sr(OH)₂/SrCl₂.” in the revised Supplementary Information.

In the main text, we add a sentence

“The molten-salt-assisted fragmentation of TaS₂ under the nitridation conditions is presumably a key reason for the formation of SrTaO₂N nanocrystals (see detailed discussion below Supplementary Fig. 9).” in the Section “Synthesis and characterization of ATaO₂N (A = Sr, Ca, Ba) nanocrystals” of the main text.

Comment 3. Following question 2, is it possible to prepare the SrTaO₂N with different particle sizes to study the effect of the particle size on photocatalytic water splitting?

Response: According to the comparison between the materials in Figure 1c, Supplementary Fig. 8 and Supplementary Fig. 11, it is possible to prepare the SrTaO₂N with different particle sizes by modulating the type/amount of the Sr source (e.g., Sr(OH)₂, SrCO₃, etc) and molten salts. However, a change in

particle size will generally induce a change in other properties, and a lot of optimizations will be needed for comparison. It is not beneficial to the research community to spend too much effort in this direction and delay the disclosure of the newly discovered unique approach to producing the functional material with small particle sizes and thus potentially more active materials.

Comment 4. As reported, by using SrCl_2 as the molten salt, the alumina crucible could be used as the aluminum source for element doping. In this work, what is the amount of Al in the prepared SrTaO_2N and the effect of doped Al for photocatalytic water splitting activity?

Response: We note that Al was not detected by XPS. This is different from the reported SrTiO_3 , due likely to the use of a new type of alumina crucibles and the much milder heating conditions (1223 K/3 h for SrTaO_2N vs. 1423 K/10 h for SrTiO_3). The lower temperature would be more critical for the lack of Al leaching from the alumina crucible.

To further answer your question, we prepared Al-doped SrTaO_2N ($\text{SrTaO}_2\text{N}:\text{Al}$, feeding Al/Ta molar ratio = 1%) nanocrystals by the developed approach using $\text{Al}(\text{NO}_3)_3 \cdot 9\text{H}_2\text{O}$ as the dopant source. The $\text{SrTaO}_2\text{N}:\text{Al}$ exhibited almost identical XRD (Figure R3a) and UV-vis DR (Figure R3b) patterns compared with the undoped specimen, whilst its background absorption intensity (Figure R3c) was slightly stronger. The $\text{SrTaO}_2\text{N}:\text{Al}$ was less active than the undoped SrTaO_2N for H_2 evolution (Figure R3d). This indicates that Al(III) is not a suitable dopant for the SrTaO_2N nanocrystals. Nevertheless, it would be interesting to investigate other aliovalent dopants (such as Mg(II), Zr(IV), Ga(III), *etc*) for further improving the photocatalytic performance in the future. We will not show the $\text{SrTaO}_2\text{N}:\text{Al}$ data in the revised manuscript to avoid unnecessary distraction from the discussion.

Figure R3. (a) XRD patterns and (b and c) UV-vis DR spectra of SrTaO₂N:Al and SrTaO₂N nanocrystals. (d) Time courses for photocatalytic H₂ evolution over SrTaO₂N:Al and SrTaO₂N nanocrystals modified with the same Cr₂O₃/Pt (MW_{EG})/Ir (MW_{H₂O}) (Cr, 0.5 wt%; Pt, 1 wt%; Ir, 0.5 wt%) cocatalysts in an aqueous methanol solution under visible light ($\lambda \geq 420$ nm). **(For Review Only)**

Comment 5. The authors attribute the high photocatalytic water splitting activity of the prepared SrTaO₂N to overcoming the long distances for the photoexcited charge carriers to migrate to reach active sites on the surface of common ATaO₂N. To confirm this conclusion, the photophysical properties of samples should be discussed by using some proper methods such as fluorescence spectroscopy, surface photovoltage intensity, and transient absorption spectroscopy.

Response: We note that the comprehensive comparison between the

developed SrTaO₂N nanocrystals and the larger-sized SrTaO₂N crystals (Supplementary Fig. 8 and 14) and nanosized polycrystalline SrTaO₂N (Supplementary Fig. 15) have clearly demonstrated the importance of both a small particle size and a high degree of crystallinity to the photocatalytic performance (please see detailed discussion in the first paragraph of the Section “H₂-evolution and Z-scheme water-splitting performance of the SrTaO₂N-nanocrystal photocatalyst”).

Surface photovoltage is not accessible for us in a reasonable time. Photoluminescence or transient absorption spectroscopy in the presence and absence of cocatalysts and reactants (i.e., pristine, loaded with cocatalysts, loaded with cocatalysts and in the presence of reactants) may be useful. The latter, which can study the active excited carriers, will be preferable and more accessible to us, but it simply takes too much time. Considering the timeliness of the newly discovered unique approach to producing the functional nanocrystal-based photocatalyst materials and thus potentially more active materials in related fields, we will not perform additional spectroscopic studies in the present work but will do it in a follow-up study.

Comment 6. How about the one-step-excitation overall water splitting activity of the prepared SrTaO₂N?

Response: The answer has already been described in the rebuttal letter of the previous version. OWS is a very demanding reaction, requiring a delicate balance of charge separation, reduction and oxidation capabilities, and thus an extensive study of particle/interface structures and compositions. One-step-excitation OWS was not yet achieved in the present SrTaO₂N nanocrystals. The inability is most likely due to the unsatisfactory surface property and/or lack of OWS-active cocatalysts (see discussion in the Section “O₂-evolution performance of CoO_x-modified SrTaO₂N nanocrystals”). It is thus our next target to enable efficient one-step-excitation OWS over these high-quality

nanocrystals of $ATaO_2N$ ($A = Sr, Ca, Ba$). Nevertheless, the progress of this work in improving photocatalytic performance of tantalum-based oxynitrides should not be underestimated and, more importantly, it would lay an important foundation and point out a clear direction for further designing long-wavelength-responsive OWS-active $ATaO_2N$ ($A = Sr, Ca, Ba$) photocatalysts.

Reviewer #3

Comment. The authors have satisfactorily replied to my concern and I believe the manuscript is suitable for *Nature Communications*.

Response: We sincerely thank the reviewer for his/her previous constructive comments/suggestions for improving the quality of our work.

REVIEWERS' COMMENTS

Reviewer #2 (Remarks to the Author):

The revision is satisfactory and I would like to recommend its publication now.